# Revealing pseudorotation and ring-opening reactions in colloidal organic molecules

P. J. M. Swinkels [1], S. G. Stuij[1], Z. Gong[2], H. Jonas[3], N. Ruffino[1], B. van der Linden[1], P. G. Bolhuis[3], S. Sacanna [2], S. Woutersen[3] & P. Schall [1✉]

Colloids have a rich history of being used as 'big atoms' mimicking real atoms to study crystallization, gelation and the glass transition of condensed matter. Emulating the dynamics of molecules, however, has remained elusive. Recent advances in colloid chemistry allow patchy particles to be synthesized with accurate control over shape, functionality and coordination number. Here, we show that colloidal alkanes, specifically colloidal cyclo-pentane, assembled from tetrameric patchy particles by critical Casimir forces undergo the same chemical transformations as their atomic counterparts, allowing their dynamics to be studied in real time. We directly observe transitions between chair and twist conformations in colloidal cyclopentane, and we elucidate the interplay of bond bending strain and entropy in the molecular transition states and ring-opening reactions. These results open the door to investigate complex molecular kinetics and molecular reactions in the high-temperature classical limit, in which the colloidal analogue becomes a good model.

[1] Institute of Physics, University of Amsterdam, Amsterdam, The Netherlands. [2] Molecular Design Institute, Department of Chemistry, New York University, New York, NY, USA. [3] Van 't Hoff Institute for Molecular Sciences, University of Amsterdam, Amsterdam, The Netherlands. ✉email: P.Schall@uva.nl

Colloidal particles have been much used to study phase transitions, such as crystallization, gelation, and the glass transition of condensed matter at experimentally accessible length and time scales[1]. In this analogy, atoms are represented by homogeneous colloidal spheres with tunable interactions between them. This approach has been successful, and many of the lessons learnt are directly applicable to fields, such as photonics[2], opto-electronics[3], and bio-materials[4].

Compared to these crystalline systems, molecules typically have a much richer structure: atoms are generally bonded directionally into molecules via valence. For example, in aliphatic organic compounds, in which a carbon atom has four bonds, such as methane ($CH_4$), the carbon atom is in an $sp^3$ hybridized state, binding ligands in a tetrahedral arrangement at ~109.5° bond angles. However, although many theoretical and simulation studies exist, achieving directional bonding in colloidal systems is challenging, and reliable experimental systems are scarce[5–7]. Recently developed patchy particles, decorated with patches of specific surface chemistry, with well-defined symmetry, allow reproducing the geometry of valence bonds[8–13]. While these particles mimic the rigid shape of small molecules, emulating molecular reactions and conformational dynamics would require degrees of freedom similar to that of organic molecules, through reversible, specific patch–patch interactions, tunable on the scale of $k_BT$, the thermal energy[11]. Such interactions would open the door to colloidal molecular chemistry: molecules that consist of colloidal particles instead of atoms, reacting at the nano and micrometre scale, directly observable by microscopy. This in turn could unlock design paths for nanostructured materials[14].

Specific, adjustable colloidal interactions arise in binary solvents close to their critical point. This so-called 'critical Casimir force' results from the confinement of solvent fluctuations between particle surfaces in a near-critical mixture, giving rise to an effective interaction that is precisely tunable with temperature and the adsorption preferences of the confining surfaces. For isotropic particles, these interactions have provided insight into gas–liquid and liquid–solid phase transitions[15,16], as well as gelation[17]. Combined with 'patchy particles' exhibiting surface patches of specific solvent affinity, the interactions become directed, and may allow the reversible assembly of more complex, molecule-like structures, and their direct particle-scale observation.

Here, we show that patchy particles bonded via critical Casimir forces allow direct-space investigation of molecular dynamics, providing insight into thermally activated molecular transition states. We bind dimer and tetramer particles into colloidal alkanes, resembling alkane molecules, such as colloidal (cyclo) butane, butyne, cyclopentane, and cyclohexane, and investigate their dynamics directly in real space, using confocal microscopy. We find that just as molecular cyclopentane, colloidal cyclopentane exhibits envelope and twist conformations that interconvert in time, and we follow the kinetic pathway in great detail. We measure the free energy and corresponding bending energy directly from the observed molecular trajectory; this allows us to elucidate the interplay of bond strain and entropy in thermal and catalytic dissociation reactions of colloidal cyclopentane. These results open up directions for studying the dynamics of molecules, using precisely coordinated patchy particles, and for building complex nano- and micrometre-scale materials[1].

## Results

**Colloid fabrication.** We fabricate patchy particles from polystyrene (PS) and 3-(trimethoxysilyl)propyl methacrylate (TPM) spheres by colloidal fusion[12]. The synthesis yields particles with precisely linearly and tetrahedrally coordinated, fluorescently labelled TPM patches in a PS matrix (Fig. 1A). The particles have a diameter of $d = 3.7$ μm and a patch diameter $d_p = 0.5$ μm; latter is sufficiently small to allow only a single other patch to bind. Di- and tetrapatch particles with number ratio 1:3 are dispersed at a volume fraction of $\phi = 0.01$ in a binary mixture of lutidine and water close to its critical point. We also add 1 mM $MgSO_4$ to screen the electrostatic repulsion and enhance the lutidine adsorption of the patches, see Supplementary Note 1. Upon approaching the solvent demixing temperature $T_{cx} = 33.8$ °C, lutidine-rich fluctuations arise between the patches, binding them into a covalent bond analogue by critical Casimir interactions, as illustrated in Fig. 1B. To assemble colloidal molecules, we heat the sample to $\Delta T = 0.04$ K below $T_{cx}$, resulting in a predicted binding energy of the patches of ~$15k_BT$ (see Supplementary Note 5)[18].

**Colloidal alkanes.** After a few hours, we observe bonded structures exhibiting the coordination and bond angles of carbon atoms in organic molecules, as shown in Fig. 1C–G (see Supplementary Note 2 for assembly details). A zigzag chain of four tetrapatch particles exhibiting the carbon backbone structure of butane is shown in Fig. 1D. A colloidal analogue of 2-butyne, consisting of two central dipatch particles with two tetrapatch particles capping both ends, is shown in Fig. 1E. Analogues of cyclic alkanes, ubiquitous in carbon chemistry, are shown in Fig. 1E–G. Rings consisting of four, five, and six tetrapatch particles are observed, which we identify as colloidal cyclobutane, cyclopentane, and cyclohexane. We note that unlike their atomic counterpart, these colloidal molecules have unsaturated patches, lacking their 'hydrogen atoms', which may lead to some difference in their reactivity and conformations.

We show size distributions of colloidal molecules in Fig. 1H. Clearly, smaller structures are in the majority, but a significant population of larger structures is present in the sample. The population decreases exponentially with size, as predicted for patchy particle systems[19]. Cyclic molecules, however, show a clear preference for a certain number of particles, reflecting their compatibility with the tetrahedral bond angle. Colloidal cyclobutane (Fig. 1F) is rarely encountered in our samples. This is not surprising considering its highly strained bond angles. In this configuration, two bonded neighbours make an angle of 90°, far from the ideal angle of 109.5°, causing high bond strain. For atomic cyclobutane, this high bond strain is known to cause the enhanced reactivity of cyclobutane compared to butane, making it much less stable than cyclopentane and cyclohexane that exhibit bond angles much closer to 109.5°. Indeed, we find that colloidal cyclopentane (Fig. 1E) is much more ubiquitous in the sample, and by far the most observed colloidal ring structure. Its bonds are much closer to the ideal 109.5° tetrahedral bond angle, compared to cyclobutane. Curiously, six-membered rings— cyclohexanes—are much less frequently observed, even though they have a lower bond angle strain, probably due to kinetic effects[7].

**Conformations of colloidal cyclopentane.** We focus on the most frequently observed cyclic structure, colloidal cyclopentane, and image its three-dimensional bonding arrangement, using confocal microscopy. The three-dimensional reconstructions reveal three basic conformations (Fig. 2A): a planar conformation with all particles in the same plane, an 'envelope' conformation, with one particle sticking out, and a 'twist' conformation with one particle sticking out above, and an adjacent particle sticking out below the plane. Just as its colloidal counterpart, atomic cyclopentane also shows these conformations; like most cyclic molecules, the internal angles of the ring are not compatible with the 109.5° bonding angle of the tetrahedral symmetry. In planar cyclopentane, the angles are only slightly lower, at 108°; nevertheless,

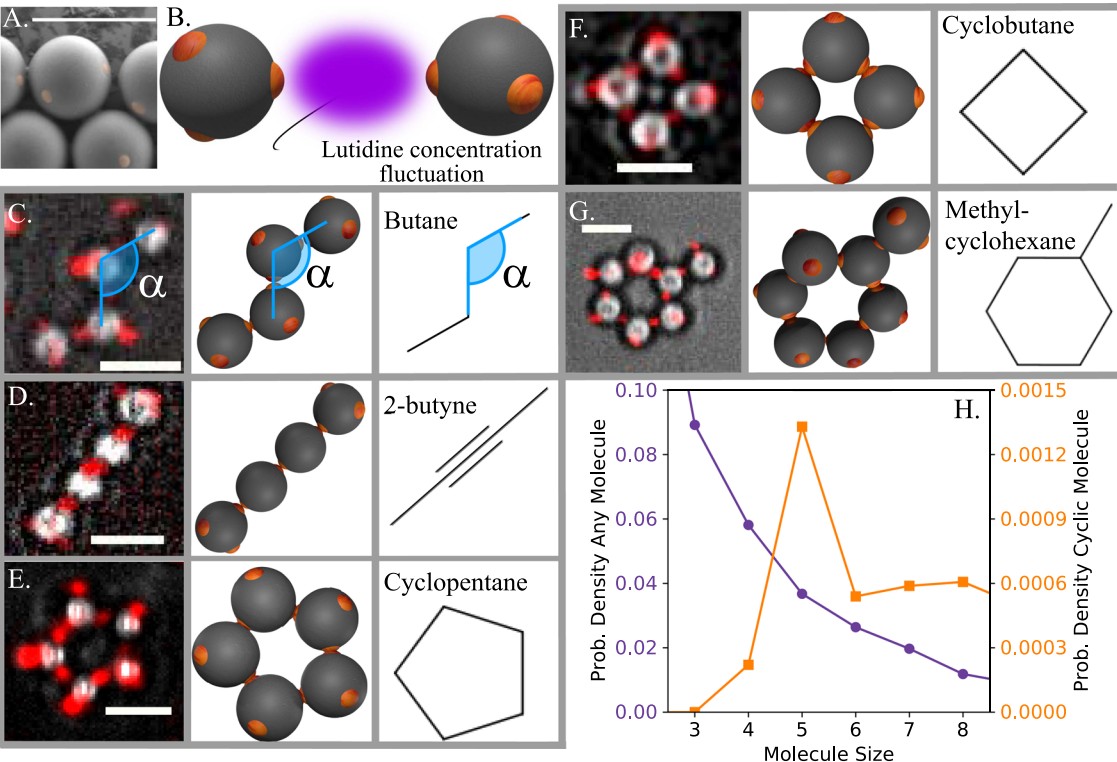

**Fig. 1 Colloidal molecules assembled by critical Casimir forces. A** Scanning electron microscope (SEM) image of tetramer patchy particle with patches coloured yellow. Scale bar corresponds to 5 μm. **B** Schematic of the directed critical Casimir interaction: lutidine fluctuations confined between hydrophobic patches cause patch-to-patch attraction. **C–G** Examples of colloidal molecules: colloidal butane (**C**), 2-butyne (**D**), cyclopentane (**E**), cyclobutane (**F**), and methylcyclohexane (**G**), confocal superimposed onto bright-field microscope images highlighting patches in red (left), schematic reconstruction (middle) and chemical representation (right). In **C**, the characteristic tetrahedral bond angle $\alpha = 109.5°$ is indicated. In **D**, the linearly connecting dipatch particles mimic *sp* hybridized carbon atoms, which in the colloidal analogue link both dipatch and tetrapatch particles. **H** Probability of occurrence of colloidal molecules as a function of their size. Purple circles (left axis) show all molecules, while orange squares (right axis) show cyclic molecules only. Clearly, larger molecules are increasingly rare, while among the cyclic structures, five-particle compounds, such as cyclopentane, are most frequently observed. Data were gathered from 163,178 clusters in 457 frames. Source data are provided as a Source data file.

due to energetic considerations—the steric hindrance of hydrogen atoms and torsional strain—atomic cyclopentane is virtually always puckered out of plane[20].

Conversely, colloidal cyclobutane shows virtually no puckering, as shown by the significantly narrower bond angle distributions in Fig. 2B. Cyclobutane shows a much narrower bond angle distribution, in particular lacking the tail towards larger angles as observed for cyclopentane, which is a signature of its puckering configurations (see Supplementary Note 3 for a comparison of bond angle distributions of different colloidal molecules). Hence, while there is a driving force for out-of plane movement of cyclopentane, and thus a relatively wide angular distribution, cyclobutane bonds are stiffer, forcing the highly strained ring into a narrowly confined configuration (see Supplementary Movie 1).

**Pseudorotation**. For each particle in cyclopentane, there are two envelope and two twist conformations, as shown in Fig. 2C. In molecular cyclopentane, these conformations are thought to interconvert continuously. This process is known as pseudo-otation, and has been suggested as early as the 40s, but so far has only been confirmed by indirect spectroscopic evidence[21–24]. Likewise, we find that in colloidal cyclopentane, envelope and twist conformations interconvert continuously. To follow the pseudorotation directly in real space, we rapidly acquire image stacks every 12 s, roughly two times faster than the typical relaxation time of a configuration, see Supplementary Note 4. Three-dimensional reconstructions reveal the pseudorotation of

colloidal cyclopentane in Fig. 2D, and Supplementary Movies 1 and 2. From frame 1 to 2, a particle (left) flips from below to above the plane, while its in-plane neighbour moves down out of plane. From frame 3 to 4, a characteristic change from envelope to a neighbouring twist conformation occurs, corresponding to the upward motion of the (central) in-plane particle. Remarkably, during these transition states, the ring remains always highly puckered. To show this, we correlate the vertical displacements, $z_i$, of nearest- and second-nearest-neighbour particles, and find that they are strongly anticorrelated, as shown in Fig. 2E: the upward movement of a particle typically results in the downward movement of the neighbouring particle, followed by an upward movement of the next-nearest neighbour. These correlations highlight the strongly correlated nature of the pseudorotation of colloidal cyclopentane, and suggest that interesting collective behaviour is to be expected in larger colloidal molecules.

To analyse the pseudorotation in detail, we determine the puckering amplitude $q$ and phase $\phi$ from the out-of-plane displacements $z_i$ of the particles. Together, $q$ and $\phi$ form a polar phase space describing all possible puckering conformations. Given an average plane through the ring, $q$ is a measure of the resulting amplitude of the out-of-plane displacements, while $\phi$ tells us in what conformation the ring is, as indicated schematically in Fig. 2C[25] (see 'Methods' for formal definitions of $q$ and $\phi$). A polar plot of the pseudorotation trajectory in $(q, \phi)$ space is shown in Fig. 2C (see Supplementary Movies 3 and 4 for animations). The coordinated up and down motion leads to pronounced changes in the puckering phase (phase change from

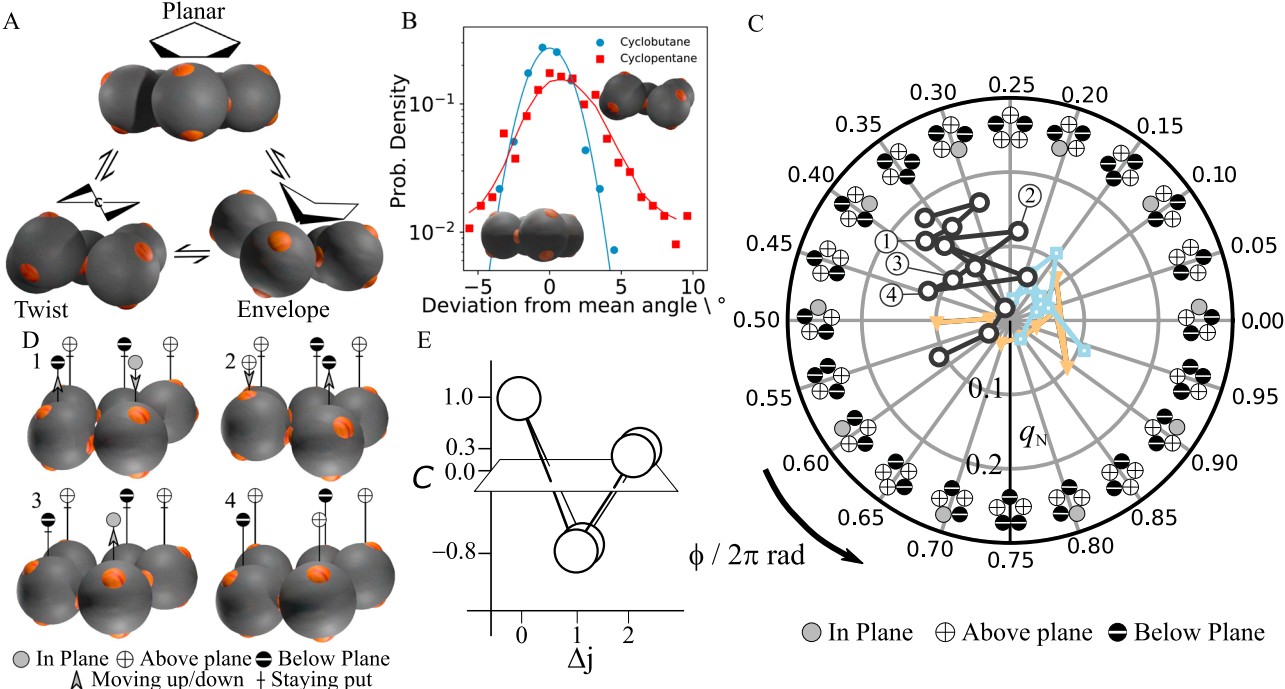

**Fig. 2 Conformations and pseudorotation of colloidal cyclopentane. A** Three-dimensional reconstructions of typical conformations of colloidal cyclopentane: planar conformation (top), 'twist' (or 'half-chair') conformation (left), and 'envelope' (or 'bend') conformation (right). **B** The distribution of inter-particle angles in cylcobutane (blue circles) and cyclopentane (red squares). The solid lines are a guide to the eye. **C** Representation of cyclopentane conformations in polar $q$–$\phi$ space. At each $0.05 \cdot 2\pi$ interval, a 2D representation of the ring is shown. The three paths (grey circles, orange triangles, and blue squares) show typical puckering routes of a thermally activated ring through $q$–$\phi$ space. The four labelled subsequent points along the grey circle path correspond to the snapshots in **D**. **D** Time series of three-dimensional configurations showing pseudorotation of colloidal cyclopentane. Snapshots are $\Delta t = 12\,\mathrm{s}$ apart, while the typical relaxation time of a conformation is 24 s, see Supplementary Note 4. Symbols + and − indicate particles above and below the average plane, respectively, and arrows indicate particle movement towards the next time step. **E** Correlation function of out-of-plane displacements $C(\Delta j) = \langle z_j \cdot z_{j+\Delta j}\rangle / \langle z_j^2 \rangle$ in a perspective view, illustrating average puckered configuration of the ring. Source data are provided as a Source data file.

frame 1–2 to 3–4). For example, the large phase change from 2 to 3 corresponds to a change from twist to a next-nearest envelope conformation, while that from 3 to 4 corresponds to a transition from twist to envelope, as shown by comparison of the trace in Fig. 2C with the corresponding 2D representations.

To explore the full statistics, we follow more rings, and determine distributions of the reduced puckering phase $\phi_r$ and amplitude $q$ as shown in Fig. 3A, B. The reduced puckering phase indicates how far a particular configuration is between an envelope ($\phi_r = 0$) and a twist ($\phi_r = 0.05$) conformation. The flat distribution of $\phi_r$ indicates that, just like in atomic cyclopentane, there is no preference for either envelope or twist conformation, nor any conformation in between. This is different for the puckering amplitude $q$ (Fig. 3B): planar conformations with $q \sim 0$ are almost never observed, while mildly puckered configurations with $q \sim 0.06$ are most prevalent. As shown by simulations in Supplementary Note 5, the peak position depends on the presence of gravity: without gravity, the maximum of the probability density is shifted towards larger values ($q_N \approx 0.15$) compared to the experimental measurement, meaning that the colloidal cyclopentane ring is more puckered in a system without gravity, as expected. Nevertheless, the presence of a finite puckering amplitude is surprising from an energetic point of view, as the flat ring ($q = 0$) has both the lowest bending and gravitational energy.

However, the entropy of this state is also the lowest: there is only one way to place the particles into a flat ring; even a small amount of puckering will unlock many configurational microstates, thus increasing the entropy.

To estimate the entropic contribution, we consider a ring of five freely joint particles moving up and down independently. The

corresponding probability distribution $P(q)$ increases linearly with $q$ (Fig. 3B), reflecting the increasing number of configurations. The experimental distribution initially follows this trend, but then peaks and diminishes. We associate this decrease with the increasing bending energy cost $U(q)$ suppressing high-puckering amplitude configurations.

Together, entropic and bending energy contributions give rise to the free energy $F(q) = U(q) - TS(q)$, where $S(q) = k_B T \ln P(q)$ is the entropy associated with the ring configurations. In thermal equilibrium, puckering amplitudes follow a Boltzmann distribution $f(q) \propto \exp(-F(q)/k_B T)$. We invert this relation to determine the average bending energy from the measured distribution $f(q)$, using $U(q) = k_B T \ln (P(q)/f(q))) + U_0$, where $U_0$ is an arbitrary reference energy, see Supplementary Note 6. The resultant $U(q)$ indeed reveals an increasing bending energy with puckering amplitude, as shown in Fig. 3D (red circles).

These trends are confirmed in Monte Carlo (MC) simulations of five tetrapatch particles interacting with effective critical Casimir potentials, as detailed in Supplementary Note 5. Their puckering distributions are in excellent agreement with the experimental data, as shown in Fig. 3. Furthermore, we compute the total bending energy $U$ directly from the strained bonds for each observed configuration; the resulting probability contour plot shows good agreement with the experimental data (Fig. 3D).

Interestingly, atomic cyclopentane shows a distribution peaked at much higher values of $q$. Contrary to colloidal cyclopentane, the molecule experiences additional H–H steric repulsions, as well as torsional strain, leading to a stronger degree of puckering. Nevertheless, the subsequent decline indicates occurrence of additional bending energy, similar to the colloidal analogue.

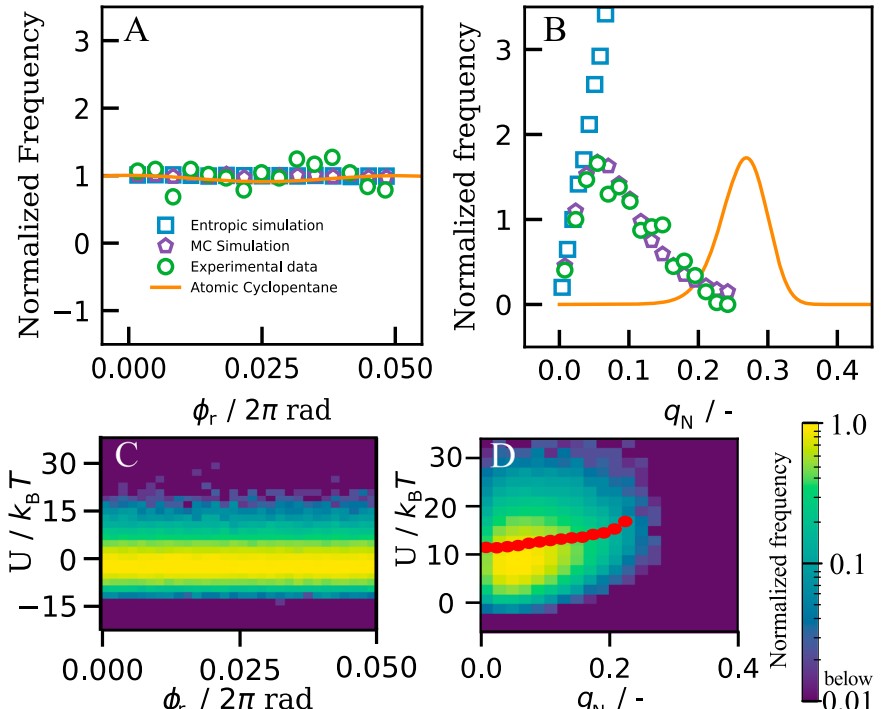

**Fig. 3 Distributions of puckering configurations. A** Normalized frequency of the reduced puckering phase angle, $\phi_r$ from envelope ($\phi_r = 0$) to twist conformation ($\phi_r = 0.05 \cdot 2\pi$ rad). The flat distribution reveals no preference for a particular conformation. **B** Frequency $f(q_N)$ of the normalized puckering amplitude $q_N$: experimental data (green circles), Monte Carlo simulations of freely joint particles (blue squares), and particles interacting with an effective critical Casimir potential (purple pentagons), and expected values for atomic cyclopentane as given in ref. [23], assuming the largest reasonable envelope-to-twist energy barrier (solid orange line). Frequency is normalized to result in identical initial slopes. **C**, **D** Probability density maps of total bending energy as a function of $\phi_r$ and $q_N$, respectively, for the simulated patchy particle rings. The logarithmic colourmap indicates the normalized frequency of observation from <0.01 (blue) to 1.0 (yellow). Red circles indicate the average bending energy as determined from the measured distribution of $q_N$. Source data are provided as a Source data file.

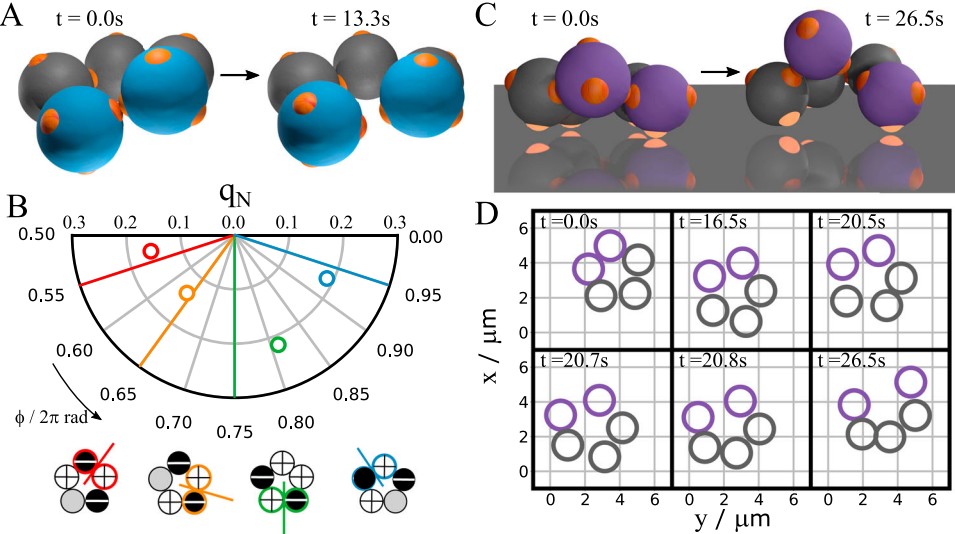

**Fig. 4 Colloidal cyclopentane ring-opening reaction. A**, **B** Thermal ring-opening of free colloidal cyclopentane. **A** Three-dimensional reconstruction of a dissociating ring just before (left) and just after breaking the bond between the blue particles (right). Particle diameter is 3.7 μm. **B** Top: examples of ring configurations just before breaking (circles) and corresponding breaking points (radial lines) in $q$–$\phi$ space. Bottom: breaking points in schematic representation, with colours indicating braking position. Colour coding matches the $q$–$\phi$ coordinates above. **C**, **D** Catalytic ring-opening of colloidal cyclopentane on an attractive (hydrophobic) surface. **C** Three-dimensional reconstruction of the ring shown before (left) and after breaking the bond between the purple particles (right). The bright patches are attached to the surface. **D** Time sequence of the surface diffusion and eventual breaking of the bond between the purple particles of the surface-bound ring.

**Ring-breaking and catalytic reactions**. We fully exploit the direct observation of colloidal cyclopentane by studying ring-opening reactions. Rings occasionally break up as shown in Fig. 4A and Supplementary Movie 5; this break up is the thermal ring-opening reaction of colloidal cyclopentane to pentane (formally a pentane bi-radical). Interestingly, we find that just before breaking, rings always exhibit a high-puckering amplitude, as shown in Fig. 4B. Moreover, the bond breaking occurs at a particle strongly involved in the puckering, as shown by comparing the puckering phase just before bond breaking with the breaking point in Fig. 4B. Because a high-puckering amplitude corresponds to high bond strain, a large energy gain is made by breaking the bond. These results highlight the importance of bond-bending strain in the thermal dissociation of colloidal cyclopentane. Furthermore, the high-$q$ values just before breaking suggest that the reactivity of cyclopentane is highest for extreme puckering amplitudes, in the far high-$q$ tail of the distribution $i$ (Fig. 3B). We speculate that a similar mechanism may apply to atomic cyclopentane. Indeed, thermal ring-breaking reactions typically occur in cyclopentane and methyl-cyclopentane[26], which unlike higher-symmetry cyclic compounds like cyclohexane exhibit significant bond strain.

For atomic cyclopentane, a catalyst can greatly accelerate the ring-opening reaction by offering attractive binding sites. For example, noble metal surfaces and mesoporous materials can achieve some selectivity in the ring-opening reaction[27–30], but the mechanism is complex and poorly understood. To investigate such catalytic reactions in colloidal cyclopentane, we assemble rings in the bulk and sediment them onto a hydrophobic surface to which the particle patches bind. We find that this greatly speeds up the ring dissociation: a few minutes after surface adsorption, a ring breaks and opens, as shown in Fig. 4C, D. Upon adsorption, three to four particle patches bind to the substrate (highlighted in Fig. 4C), confining the ring to the surface, and effectively freezing it in a single envelope conformation. At the same time, the ring can still move laterally and diffuse along the substrate, before it opens (Fig. 4D). The binding-induced angular strain, together with the entropically unfavourable locking of configurations, causes the ring to break easily. The hydrophobically treated substrate serving as a catalyst thus allows direct observation of the interplay of conformations, bond geometry and surface interaction in the catalytic dissociation process. This makes our colloidal model very suitable for the study of geometric effects of catalysts, in absence of any electronic or support effects, using designed colloidal crystal surfaces or templates with defined symmetry, lattice constant, and domain size. Fast confocal microscopy imaging can then give detailed insight into energies along the reaction coordinate, identifying the transition states in these catalytic processes.

## Discussion

The direct observation of pseudorotation and catalytic reactions of colloidal cyclopentane offers insight into transition states in a molecular dissociation reaction. As the thermal energy $k_BT$ is of the same order as the bond energy, the colloidal molecule corresponds to its high-temperature molecular counterpart. The ratio of 15 for the bond energy to $k_BT$ in our experiments would correspond to a temperature of ~640 K for atomic cyclopentane, as can be estimated from its C–C bond energy of ~80 kJ/mol. In this limit, the quantum mechanical energy spectrum of the low-frequency modes of the molecule becomes quasi-continuous, and classical behaviour is expected to emerge, suggesting that the colloidal analogue becomes a good model.

Our results highlight the importance of bond strain in the dissociation process, and suggest specific catalyst design that takes advantage of the puckering amplitude in the ring-breaking process. These results pave the way to the investigation of molecular kinetics by detailed direct observation of colloidal analogues, elucidating transition states, kinetic pathways, and correlations in molecular reactions[31]. The exquisite temperature control afforded by the critical Casimir interaction creates opportunities for molecular and supramolecular colloidal design, following equilibrium and non-equilibrium routes, and the investigation of dynamic assembly pathways. The demonstrated accurate binding control opens up pathways to 'colloidal molecular chemistry', in which bond-stretch and -bend potential-energy functions can be tuned by the experimentalist, and all the reactions, not only ring opening but also synthesis, homogeneous catalysis, and polymerization can be followed in real time, using a conventional microscope to observe the reacting 'colloidal atoms'.

## Methods

**Particle synthesis**. Patchy particles are made via colloidal fusion[12] with a modified recipe. Briefly, negatively charged PS particles are synthesized by seeded emulsion polymerization, using potassium persulfate (>99.0%, Sigma-Aldrich) as initiator. After repeated seeded emulsion polymerization, negatively charged PS particles with a diameter of ~2.2 μm are resuspended into 12.5 mM NaCl solution. Silicone emulsion droplets with a diameter of 1.0 μm are made by hydrolysis and condensation of TPM (98%, Sigma-Aldrich) catalysed by ammonium hydroxide (28 wt.%, Sigma-Aldrich). The droplets are later fluorescently labelled with 3-aminopropyl trimethoxysilane-coupled rhodamine-B isothiocyanate. Pluronic F108 triblock copolymer (average Mn 14,600, Sigma-Aldrich) is added into the silicone emulsion to a final concentration of 0.05 wt.% to further stabilize the droplets. The F108-TPM emulsion is gently washed by two cycles of centrifugation/resuspension to remove free F108 polymer in emulsion, and later resuspended into 12.5 mM NaCl solution. Tetrahedral colloidal clusters are made by mixing the prepared negatively charged PS particles of 2.2 μm and F108-TPM droplets of 1.0 μm at a number ratio ~100:1. To screen the negative charge on both particle surfaces and thus allow for polymer bridging between the two, 12.5 mM NaCl is necessary. These clusters are then separated from excess PS singlets by centrifugation in glycerol/water mixture (~22 wt.% glycerol). The purified clusters are resuspended into aqueous solution containing 0.05 wt.% F108 copolymer and 0.5 wt.% dodecyltrimethylammonium bromide (≥98%, Sigma-Aldrich), to prevent aggregation, and facilitate core extrusion during colloidal fusion, separately. Pure tetrahydrofuran is then added to cluster suspension to reach a final concentration of 30% $V_{THF}/V_{cluster}$. The fusion is allowed to proceed for 3 min before being quenched by deionized water. The resulting patchy particles with liquid patches are then resuspended into deionized water, followed by its polymerization in 80 °C oven for 2 h. After polymerization, patchy particles typically undergo three centrifugation/resuspension cycles and transferred back into deionized water for further experiments. Dipatch particles are the fusion product of trimer colloidal clusters, which are assembled and fused by the same protocol described above, except the TPM emulsion droplets used to make clusters are smaller (diameter being ~0.6 μm).

The patchy particle dimensions were determined using atomic force microscopy (AFM; Supplementary Table 1 and Supplementary Fig. 1) and confirmed by observation of inter-particle distances of bonded tetramer particles (Supplementary Fig. 2). AFM was performed on dried particles on a glass slide. We locate particles where the protruding patch is pointing upwards, so they are suitable for analysis. Very good consistency is observed: the mean of the inter-particle distance distribution at 3.8 μm reflects the AFM-determined particle size (particle diameter 3.7 μm) plus twice the patch height (2 × 48 nm = 96 nm), plus the (short) interaction range, while the standard deviation of $\sigma = 0.11$ μm reflects the estimated particle size variation (100 nm) and twice the variation of the patch height (2 × 5 nm = 10 nm).

**Sample details and critical Casimir interactions**. A mixture of di- and tetrapatch particles (of number ratio 1:3) is dispersed in a binary solvent of 25% 2,6-lutidine (≥99.0%, Sigma-Aldrich) and 75% MilliQ water with 1 mM $MgSO_4$ (≥99.5%, Sigma-Aldrich). The particles are washed several times in the water–lutidine mixture. The resulting particle dispersion is injected into a glass capillary (Vitrotubes, Rectangle Boro Tubing 0.20 × 2.00 mm) and sealed with teflon grease (Krytox GPL-205).

**Confocal microscopy and particle tracking**. Particles are left to sediment to the bottom of the sample at room temperature before measurements. We choose a particle concentration such that particles cover between 10 and 15% of the surface after sedimentation. We then switch on the critical Casimir attraction to assemble the structures.

We use a well-controlled temperature stage in combination with an objective heating element to obtain a relative temperature accuracy of ~0.01 °C, with minimal temperature gradients.

In an experiment, we heat the sample to 0.04 °C below the phase separation temperature of the lutidine–water mixture, inducing critical Casimir attraction between patches. The structures then grow by two-dimensional diffusion in the plane. No mixing is necessary. After several hours of equilibration, we investigate the structures using a 63× oil-immersion objective.

For the catalytic experiments, we assemble the alkenes in the bulk. This is because in these experiments, the walls are treated to exhibit strong particle–wall interaction. To investigate their 'catalytic effect', we assemble the particles in the bulk, and then let the entire assembled structure sediment to the bottom, so that after contact with the wall, the catalytic process can begin. This is done by first letting the particles (without critical Casimir force) sediment on one side of the capillary, turning the capillary upside down, and then switch on the Casimir interaction so that rings form in the bulk (while sedimenting to the other side of the capillary). This process allows us to study the catalytic effect on cyclopentane rings that have assembled in the bulk.

We image the assembled structures using confocal microscope image stacks, alternating with bright-field images. To follow a colloidal molecule in time, we acquire ~100 image stacks and bright-field images during a time interval of 12 min. The bright-field images are processed using particle tracking software (Trackpy[32]) to determine the centre of the patchy particle in the horizontal plane. The 3D locations of the fluorescent features are determined using the same software. The colloidal ring structures were then detected using an image analysis algorithm, and confirmed by manual inspection of the video.

**Pseudorotation in polar coordinates**. We use the method by Cremer and Pople[25] to find the ring puckering coordinates for each particle configuration. This is done by expressing the positions $\mathbf{R}_j$ of the particles in a new coordinate system. In this new coordinate system, the average plane through all particles forms the $xy$-plane, and the $z$-displacement $z_j$ is the distance of a particle from this plane.

This new coordinate system must satisfy

$$\sum_{j=1}^{5} \mathbf{R}_j = 0 \quad \text{(origin set to center of mass)} \tag{1}$$

meaning that the origin of the system is at the centre of mass of the five particles. We further enforce

$$\sum_{j=1}^{5} z_j = 0 \quad \text{(mean plane through origin)} \tag{2}$$

which means all $z$-displacement $z_j$ in the new coordinate systems average out to zero: the mean plane must be the average of the particles. We choose the unit vector $\mathbf{l}$ going through any of the five particles as a convenient $x$-axis. Unfortunately, Eqs. (1) and (2) do not yet yield a unique average plane through the five particles: multiple planes can be drawn that will all satisfy the above conditions. Therefore, we further impose the conditions

$$\sum_{j=1}^{5} z_j \cos[2\pi(j-1)/N] = 0 \tag{3}$$

$$\sum_{j=1}^{5} z_j \sin[2\pi(j-1)/N] = 0 \tag{4}$$

which will fix the plane uniquely. In the case of a regular planar polygon, Eqs. (3) and (4) correspond to the condition that a small displacement $z_j$ will not lead to angular momentum. The same conditions may be used more generally for bigger displacements, and any bond lengths or angles. Also, this condition will yield the same plane irrespective of which particle is chosen as $j = 1$. Combining all these conditions, we can now determine the orientation of the mean plane for the position vectors $\mathbf{R}_j$:

$$\mathbf{R}' = \sum_{j=1}^{5} \mathbf{R}_j \sin(2\pi(j-1)/N) \tag{5}$$

$$\mathbf{R}'' = \sum_{j=1}^{5} \mathbf{R}_j \cos(2\pi(j-1)/N) \tag{6}$$

then the unit vector

$$\mathbf{n} = \mathbf{R}' \times \mathbf{R}'' / \left| \mathbf{R}' \times \mathbf{R}'' \right| \tag{7}$$

is perpendicular to $\mathbf{R}'$ and $\mathbf{R}''$. Thus, we can use $\mathbf{n}$ to define the new $z$-axis. Finally, the $y$-axis, defined by unit vector $\mathbf{m}$, can now be found by simply taking the crossproduct of $\mathbf{l}$ and $\mathbf{n}$. By linearly displacing coordinates to this new coordinate system, we can determine the ring's puckering coordinates.

The ring puckering amplitude $q$ and phase $\phi$ for a ring of five particles are defined as

$$q \cos \phi = \sqrt{2/5} \sum z_j \cos 4\pi j/5 \tag{8}$$

$$q \sin \phi = \sqrt{2/5} \sum z_j \sin 4\pi j/5 \tag{9}$$

We can simply calculate the puckering coordinates from the transformed coordinates for every snapshot.

**Simulations**. The MC calculations are performed with a potential constructed from the universal scaling behaviour of the critical Casimir interactions, and benchmarked onto experimentally measured static and dynamic equilibrium properties of a dipatch system at various temperatures. As the dipatch and tetra-patch particles are synthesized from the same materials, immersed in the same solvent, and differ only in their physical dimensions, their potentials are similar. The bond directionality and flexibility of the patchy particles is incorporated via the switching function $S$ (see Supplementary Note 5).

## Data availability
The datasets generated during and/or analysed during the current study are available from the corresponding author on reasonable request. Source data are provided with this paper.

## Code availability
The codes of the computer simulations and data analysis are available from the corresponding author upon request.

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

## Acknowledgements

P.S. acknowledges support by a Vici Fellowship from the Netherlands Organization for Scientific Research (NWO). P.S. and P.B. acknowledge support by grant 680.91.124 from NWO. We thank Michele Zanini for the AFM data and analysis.

## Author contributions

P.J.M.S., S.G.S., S.W., and P.S. conceived the study. P.J.M.S. performed the experiments with help of N.R. and B.vd.L., and analysed the data. H.J. and P.G.B. performed the simulations. Z.G. and S.S. made the particles. P.J.M.S. and P.S. wrote the manuscript. All authors discussed the data and reviewed the manuscript.

## Competing interests

The authors declare no competing interests.
