## [Peer Review File · Nature Communications]

Reviewers' Comments:

Reviewer #1:

Remarks to the Author:

The authors of this paper present beautiful experimental and numerical data on the realization of colloidal analogues of small organic molecules and in particular focus on the existence of the transitions between different conformations of a colloidal cyclopentane structure. The structure is particularly interesting because it shows different internal modes that are associated to an out-of-plane deformation, which makes it challenging to be experimentally captured.

I find in particular the experiments very challenging. Anyone who has ever tried to synthesis these patchy particles knows how hard it is to get the level of control shown by the authors. Moreover, and perhaps even harder, anyone who has tried to work with controlled colloidal assembly must appreciate how delicate the balance between electrostatics and critical Casimir forces must be to ensure the right depth of attraction to enable the manifestation of the internal reconfiguration of the colloidal structures shown by the authors.

Finally, the demonstration of this complex internal dynamics places itself in a broader research field, where some of the authors have made significant contributions, and will interest a broad readership.

In general, I am supportive of the publication of this manuscript, but there are several points that I believe would add further substance to the work and that should be addressed before publication.

1) The authors focus on a single structure but give no information of the internal dynamics of other, e.g., simpler structures. It would be of interest for the reader to see that similar conclusions can be reached for some of the other conformations. If the authors have the data, the manuscript would greatly benefit from a more generic description of the single bonds as a build up to show that the binding energies and the characteristic time scales for the cyclic structure can be rationalized from the single-particle properties.

2) The authors give no information on the kinetics of the formation of these structures nor on their distribution. This omission is particularly striking because they present their results as the first steps to a new generalized way to describe any chemical reaction. I encourage the authors to show quantitative data on these aspects.

In addition to these two main aspects, I have a general comment, which refers to a perhaps general tendency in the colloidal molecules/assembly community. There seems to be a drive to justify this kind of research from an applied standpoint, mentioning some generic strategy for the development of new materials (e.g. see line 45-46 "This in turn would unlock new design paths for future materials"). I frankly do not see how from such colloidal assemblies, given the time scales and the incredibly delicate assembly conditions, one can seriously think to develop robust strategies for the realization of new materials on a large enough scale to concern applications. I think that the research presented in this paper is interesting in its own right, without the need to chase the development of new materials and I would support the authors in acknowledging this.

I have also little appreciation of sweeping claims like (line 194) "This work opens up a new field of science ('colloidal molecular chemistry')...". Here I would encourage the authors to tone this down and justify the impact of their beautiful results in the framework of a long-standing effort to investigate complex colloidal assemblies.

In general, I found the paper well written and I only have two further minor comments:

- Figure 2D and E are mixed up in the figure caption. The description of Figure 2E (the transition states diagram) is not clear. There is no explanation of the different colors, nor of the numbers in the figure and a more detailed description of how to interpret the graph must be provided in the text.

- The description of the graphs in Figure 3 is very concise and it became much clearer to me after reading the Methods section. I would encourage the authors to expand it a little and perhaps move some of the material from the methods to the main text, to make the meaning of the plots immediately clear to the reader (this also partly refers to Figure 2E)

Reviewer #2:

Remarks to the Author:

This paper reports the dynamics of colloidal molecules formed by patchy particles with tetrahedral bonds based on critical Casimir forces. Using confocal microscopy, the 3-dimensional conformations of a 5-member ring (cyclopentane analog) are reconstructed at successive time points. Conformations are characterized by two parameters (q and ϕ) that describe continuous transitions among planar, twist, and envelope conformations. Experimentally observed distributions of these parameters agree with those of MC simulations that account for the critical Casimir interactions between particle patches. This system allows for real-time, real-space observation of ring-opening reactions in colloidal cyclopentane catalyzed by a hydrophobic substrate. In this way, colloidal molecules with directional bonds of tunable strength provide a classical analog by which to model chemical transformations.

The experiments, analysis, and modeling are of high quality and demonstrate remarkable control over the direction and strength of colloidal interactions. The agreement between experiments and MC simulations suggest that the system energy is well described by the particle geometry, critical Casimir interactions, and gravity. The potential for investigating catalytic reaction pathways and reactive intermediates using this experimental model is intriguing.

For these reasons, I recommend publication of this work in Nature Communications. I would appreciate it, however, if the Authors could address the following question.

1) The demonstration of heterogeneous catalysis is quite interesting. Can the Authors comment further on the reaction pathway and the associated energy landscape? Using the reconstructed configurations and the model energy function, it should be possible to construct the energy vs. reaction coordinate diagram and/or identify the transition state. An expanded discussion of this particular example could be helpful in illustrating the potential role of colloidal models in studying "the geometric effects of catalysts".

Minor point:

2) Figure 2. Captions for (E) and (D) are reversed.

Reviewer #3:

Remarks to the Author:

The authors present a model system of colloidal molecules using a novel fabrication method based on the assembly of single Polystyrene (PS) colloidal patchy particles via Casimir forces. The formation of the various colloidal alkene structures, such as butane, 2-butyne, and cyclopentane, depends on the patch number and orientation on the PS surface (e.g. dimers and tetramers) and how this interact. In particular, the authors visualize the bonding arrangements of cyclopentane using confocal microscopy and characterize the pseudorotation distinguishing the amplitude and the phase, which shows an increasing number of configurations. They support their experimental results with a simulation at the colloidal scale and comparing it with the analog molecular structure. Moreover, they also explore the possible effect of the interaction with the surface on the ring-opening as an analogy for the reactivity of such molecules in the presence of catalyzers. All the results are presented in a very clear and coherent manner. This is an interesting work that shows the vast possibilities of these new colloidal structures to be used as a model for molecular structures to elucidate transition states and specific kinetics.

However, there are some comments and issues that need to be addressed before I can recommend it for publication:

1. From a fabrication point of view, what are the final structure populations? In particular, could the authors elaborate more on the count of cyclopentane structures, as well as for the counts in events?

2. There is an inconsistency in the PS particle diameter: in the main text, the authors state that the PS pare has a diameter of 3.7 μm , while in the Materials and Methods they say 2.2 μm . This

information might be crucial, as the ratio between PS and patch size might play a crucial role during the molecule fabrication. It would also be useful to have information about the polydispersity of the patches.

3. Due to the size of the particles, I would expect sedimentation during the alkene assembly step. Could the authors give more experimental information on how the process of assembling the alkenes is done? Is stirring a key parameter to avoid sedimentation and ensure good mixing and therefore interaction of the various patchy particles?

4. From Fig. 1, it is not obvious that the distance from each patchy particle might be constant. Could the authors provide more information on this? The possible polydispersity of the patches (see comment #2), might affect the interparticle distance, and most probably in the case of cyclopentane the reconfiguration events.

5. The authors argue that the shift of the distribution of puckering amplitude to higher values of q for molecules is due to the steric repulsion of H-H. This raises a great experimental challenge, as one could aim to achieve such a steric repulsion in some way. Have the authors considered the presence of the patches somehow in the simulations?

6. The fact that $MgSO_4$ is a multivalent salt results in a higher ionic strength of the solution in comparison to using the same molarities for KCl or $CaCl_2$. I wonder if the authors could elaborate a bit more on this and on the clear effect on the colloidal molecules assembly.

7. I have my reservations towards the claim of using the surface as an analog for a catalyst. Could the authors prove that the interaction with the hydrophobic surface is strong enough to break a 15 kBT bond between the patches? What is the treatment done on the surface to achieve such hydrophobicity?

Reviewer #4:

Remarks to the Author:

I read the manuscript "Revealing pseudorotation and ring-opening reactions in colloidal organic molecules" by P. J. M. Swinkels and co-workers. The authors harness critical Casimir forces to fabricate clusters of microparticles that mimic the architecture of polyatomic molecules. The valency is implemented using colloids equipped with sticky patches, whereas the floppiness of the bonds stems from the small magnitude of the attractive forces. After showcasing few examples of 'colloidal molecules' that can be assembled using this method, the authors focus their attention on the dynamics of cyclopentane, finding strong similarities (and differences) with the puckering configurations of their molecular counterparts.

I find the results presented in the manuscript novel and interesting. As correctly pointed out in the Introduction, the literature is rich of 'colloidal atoms' but there are only few examples of colloidal molecules because directional bonds are not easy to implement. In particular, I am not aware of any existing work in which cyclic colloidal molecules have been achieved experimentally. On the other hand, I believe that the manuscript could be more transparent as important pieces of information are now hidden behind a nice story. My main remarks are listed below, in descending order of importance.

1) I find the title, abstract and introduction partly misleading and overstating. While it is claimed that "colloidal alkanes assembled from tetrameric patchy particles undergo the same chemical transformations as their atomic counterparts", this is only shown (nicely) for colloidal cyclopentane. There is no result concerning the other molecules shown in Fig. 1.

2) It is unclear to me what the authors exactly see while setting up an experiment. If I understand correctly, they prepare a suspension and the particles self-assemble into 'molecules' when they get close to contact (at $T \approx T_{\{cx\}}$). Hence, there is little control over the final cluster(s) beyond the binding sites. It is mentioned that some molecules are "rarely encountered", whereas "cyclopentane is by far the most observed colloidal ring structure". It would be useful to

understand a bit more the statistics and how it evolves. This is particularly important since the molecules have open bonds and clusters will, therefore, grow with time. On the other hand, do they ever spontaneously break?

3) Another relevant issue is the particles-substrate affinity. Near the critical point I expect the Casimir interactions between the colloids and substrate to come also into play. This effect is discussed in relation to Fig. 4, but not considered before. Why? The fact that the substrate is not functionalised (as in Fig. 4) is not enough to guarantee that critical Casimir interactions do not exist. Do the authors observe free rotational diffusion of single particles? This would indicate that these particle-substrate interactions are somewhat negligible.

4) Page 3, line 76. "... resulting in a binding energy of $\sim 15kT$ ". Where does this number come from?

5) Page 3, line 100. "We rapidly acquire image stacks every 10 seconds, roughly 2 times faster than the typical relaxation time of a configuration". Do all configurations have the same relaxation time?

6) I would spend few words to introduce the puckering amplitude and phase in the main text (not only in the Methods).

7) The puckering amplitude does not depend on gravity. This is an important piece of information, but it is only given at the end the Supplementary file.

Ivo Buttinoni

Reviewer #1 (Remarks to the Author):

The authors of this paper present beautiful experimental and numerical data on the realization of colloidal analogues of small organic molecules and in particular focus on the existence of the transitions between different conformations of a colloidal cyclopentane structure. The structure is particularly interesting because it shows different internal modes that are associated to an out-of-plane deformation, which makes it challenging to be experimentally captured.

I find in particular the experiments very challenging. Anyone who has ever tried to synthesis these patchy particles knows how hard it is to get the level of control shown by the authors. Moreover, and perhaps even harder, anyone who has tried to work with controlled colloidal assembly must appreciate how delicate the balance between electrostatics and critical Casimir forces must be to ensure the right depth of attraction to enable the manifestation of the internal reconfiguration of the colloidal structures shown by the authors.

Finally, the demonstration of this complex internal dynamics places itself in a broader research field, where some of the authors have made significant contributions, and will interest a broad readership. In general, I am supportive of the publication of this manuscript, but there are several points that I believe would add further substance to the work and that should be addressed before publication.

Thank you very much for your supportive judgement.

1) The authors focus on a single structure but give no information of the internal dynamics of other, e.g., simpler structures. It would be of interest for the reader to see that similar conclusions can be reached for some of the other conformations. If the authors have the data, the manuscript would greatly benefit from a more generic description of the single bonds as a build up to show that the binding energies and the characteristic time scales for the cyclic structure can be rationalized from the single-particle properties.

Thank you for this comment, we very much agree. Indeed, we initially started with di-patch particles forming polymer analogues, and studied in detail their internal dynamics, from single bond fluctuations to mode fluctuations of the chain, to determine its bending rigidity. These results show that we can very well understand the chain bending rigidity from the single-bond properties, which can be both tuned by temperature. This work will be published elsewhere.

Fig. A1 Distribution of bond angles of small linear and cyclic colloidal molecules

The data indicates the distribution of the angles in the horizontal plane for colloidal propane (blue dots), butane (green dots) and cyclobutene (orange dots). Black solid line indicates a Hookean fit with a spring constant of $k = 3.4k_B T / \text{rad}$.

Fig. A2 Distribution of inter-particle angles.

The data shows the distribution of the angles in 3D between adjacent tetrapatch particles in cyclobutane (blue) and cyclopentane (red). Solid lines are a guide to the eye. Clearly, the distribution of colloidal cyclobutane is much narrower than that of cyclopentane, indicating its higher internal strain making it stiffer and less prone to puckering.

For the current article, following the referee's comment, we have performed a similar bond-bending analysis on the current 4-patch particles, starting from a simple isolated three-particle bond angle (colloidal propane), via a four-particle structure of two joint bond angles (butane) to a highly strained, cyclic molecule (cyclobutane), and the more relaxed cyclic molecule (cyclopentane) studied in the paper, see Fig. A1 above. The angle distributions suggest that open molecules have similar bond stiffnesses (see colloidal propane and butane) with a typical value of $\sim 3 k_B T/\text{rad}$, while strained cyclic compounds show stiffer bonds due to additional constraints and pre-strain in the cyclic structure. Interestingly, the high bond strain of cyclobutane forces it into a flat configuration, suppressing any significant puckering, unlike cyclopentane. The puckering of the latter leads to a significantly broader angle distribution of the full three-dimensional angles as shown in Fig. A2 above, evidencing the puckering motion described in the manuscript. The bending properties and energies reported in the manuscript thus naturally derive from the properties of the single patchy bonds. This is also reflected in the comparison of the computed and experimental bending energy in the inset of Fig. 3b in the manuscript. The contour plot (simulations) shows the computed bond bending energies of the configurations, while the red line (experiments) shows the energies derived from Boltzmann distributions.

Changes to the manuscript:

We have included Fig. A1, together with corresponding text in the supporting information (new paragraph "Supplementary Note 2 Conformations of different colloidal molecules" and Supplementary Figure 4). To highlight the role of pre-strain, and elucidate its relation to the puckering of cyclic molecules, we have included Fig. A2 together with corresponding discussion in the main text (new Fig. 2c).

2) The authors give no information on the kinetics of the formation of these structures nor on their distribution. This omission is particularly striking because they present their results as the first steps to a new generalized way to describe any chemical reaction. I encourage the authors to show quantitative data on these aspects.

We have performed additional analysis on the kinetics of the assembly, see Fig. A3 below. The figure shows both real-space snapshots of the assemblies, and the evolution of a single cyclopentane molecule, as well as the evolution of the cluster size. The assembly process of a colloidal (methyl-)

Fig. A3 Kinetics of the patchy particle assembly.

A) Bright-field microscope image of tetra- and di-patch particles assembling after 2 hours. The dimensions of the image are $97\ \mu\text{m}$ by $83\ \mu\text{m}$. (B) Four bright-field microscope images showing the typical assembly of a colloidal (methyl-)cyclopentane ring. In this particular case, we start with butane and ethane, which react to form hexane. The molecule then reacts with itself to form methyl-cyclobutane. (C) Growth of the average cluster mass with time. Yellow line is the moving average of 100 snapshots. (D) Cluster-mass distributions of the sample after 2, 10, 30, 60, and 110 minutes. Inset: first three points in a lin-lin representation.

cyclopentane ring is shown in Fig. A3B, revealing how in this case the ring builds up from colloidal butane and ethane, which react to form hexane. The molecule then closes to form (methyl-)cyclopentane. Data on the global growth statistics are provided in Fig. A3C and D. The average cluster size grows continuously with time as shown in Fig. A3C. As can be seen from the cluster size distributions in Fig. A3D, this is due to an emerging population of larger structures becoming more prominent with time. Interestingly, as larger structures emerge, there appears to establish a preference for certain cluster sizes such as those consisting of $n=5$, 7, and possibly 10 (though statistics on this end are low) particles, indicating that some structures are energetically more favorable than others. This is indeed interesting for future work on a more general modelling of chemical reactions.

Changes to the manuscript:

We have included Fig. A3, together with a whole new section on assembly kinetics in the SI ("Supplementary Note 1: Patchy Particle Assembly"). Furthermore, and also in response to other comments by the referee, we have added a new panel showing cluster size distributions in Fig. 1H of the manuscript. The orange data therein clearly shows the preference for 5-particle rings in cyclic structures.

In addition to these two main aspects, I have a general comment, which refers to a perhaps general tendency in the colloidal molecules/assembly community. There seems to be a drive to justify this kind of research from an applied standpoint, mentioning some generic strategy for the development of new materials (e.g. see line 45-46 "This in turn would unlock new design paths for future materials"). I frankly do not see how from such colloidal assemblies, given the time scales and the incredibly delicate

assembly conditions, one can seriously think to develop robust strategies for the realization of new materials on a large enough scale to concern applications. I think that the research presented in this paper is interesting in its own right, without the need to chase the development of new materials and I would support the authors in acknowledging this.

We largely agree with the referee here; specifically, the cited sentence may be too broad and thus not helpful. However, we do have applications in mind, which we are already pursuing. These applications rather concern nanoscale-particles, in particular quantum dots. We have shown (Marino *et al.* J. Chem. Phys. 2019) and are currently researching the use of critical Casimir forces to assemble semiconductor nanoparticles (CdSe and lead-halide perovskites), which have useful optoelectronic applications, and because of their natural facets give rise to some directed interaction (manuscript submitted). The advantage here is that for nano-scale particles, ΔT becomes much larger, up to 5 or even 10°C to yield the same ratio of interaction range to particle size, making the effect much easier to apply. We also find that (possibly due to van der Waals forces at that scale) once assembled, the structures are fixed and remain fixed after drying. Hence, we believe this critical Casimir control could be useful for the assembly of ordered films of quantum dots for optoelectronic devices, such as LEDs and photovoltaic films, and we have already secured a patent in this direction (WO 2017109123A1), “Method for assembling semiconductor nanocrystals”). However, we agree with the referee, that the cited statement was too broad, and we have specified it and added a citation to our work to better reflect the potential applications we have in mind.

Changes to the manuscript:

The sentence has been modified to: “This in turn could unlock new design paths for nanostructured materials [14].”

I have also little appreciation of sweeping claims like (line 194) “This work opens up a new field of science (‘colloidal molecular chemistry’)...”. Here I would encourage the authors to tone this down and justify the impact of their beautiful results in the framework of a long-standing effort to investigate complex colloidal assemblies.

We agree and have adjusted this sentence to better reflect the achievement of our work with respect to the larger effort towards complex colloidal assemblies.

Changes to the manuscript:

We modified the sentence to:

“The demonstrated accurate binding control opens up new pathways to ‘colloidal molecular chemistry’, in which bond-stretch and -bend potential-energy functions can be tuned by the experimentalist, and ...”

In general, I found the paper well written and I only have two further minor comments: - Figure 2D and E are mixed up in the figure caption. The description of Figure 2E (the transition states diagram) is not clear. There is no explanation of the different colors, nor of the numbers in the figure and a more detailed description of how to interpret the graph must be provided in the text.

We thank the reviewer for noticing the mix up of panels D and E in the caption of Fig. 2. We have fixed this issue. Furthermore, following his suggestion, we have expanded on the explanation of panel E (panel F in the revised manuscript), clarifying the meaning of color and numbers, and adding some more description to guide the reader.

Changes to the manuscript:

The revised caption reads “(F) Transition states in q - ϕ space during pseudo rotation of three colloidal cyclopentane rings. The three paths (grey, orange and blue) show typical puckering routes of a thermally activated ring through q - ϕ space. The four labelled subsequent points along the grey path correspond to the snapshots in panel (D).”

- The description of the graphs in Figure 3 is very concise and it became much clearer to me after reading the Methods section. I would encourage the authors to expand it a little and perhaps move some of the material from the methods to the main text, to make the meaning of the plots immediately clear to the reader (this also partly refers to Figure 2E).

We thank the reviewer for this comment, and have incorporated his suggestion. We have expanded on the definitions of q and ϕ , and incorporated some of the material from the methods and the SI to the main text.

Changes to the manuscript

We have expanded on the explanation of puckering amplitude and phase as follows: “To analyse the pseudorotation in detail, we determine the puckering amplitude q and phase ϕ from the out-of-plane displacements z_i of the particles. Together, q and ϕ form a polar phase space describing all possible puckering conformations. Given an average plane through the ring, q is a measure of the resulting amplitude of the out-of-plane displacements, while ϕ tells us in what conformation the ring is, as indicated schematically in Fig. 2B [25]. (see Methods for formal definitions of q and ϕ).”

Furthermore, to better explain the position of the maximum in Fig. 3B (and also in reply to referee 4), we have added some simulation result from the SI, according to: “As shown by simulations in Supplementary Note 4, the peak position depends on the presence of gravity: without gravity, the maximum of the probability density is shifted towards larger values ($q_N \sim 0.15$) compared to the experimental measurement, meaning that the colloidal cyclopentane ring is more puckered in a system without gravity, as expected. Nevertheless, the presence of a finite puckering amplitude is surprising from an energetic point of view, as the flat ring ($q = 0$) has both the lowest bending and gravitational energy.”

Finally, to better explain the transitions in Fig. 2E (2F in the revised manuscript), we have added reference to the schematic in Fig. 2B. We added: “For example, the large phase change from 2 to 3 corresponds to a change from twist to a next-nearest envelope conformation, while that from 3 to 4 corresponds to a transition from twist to envelope, as shown by comparison of the points in Fig. 2F with the corresponding phase angles in panel (B), from twist (T-1) to envelope (E+2) to twist (T-4).”

Reviewer #2 (Remarks to the Author):

This paper reports the dynamics of colloidal molecules formed by patchy particles with tetrahedral bonds based on critical Casimir forces. Using confocal microscopy, the 3-dimensional conformations of a 5-member ring (cyclopentane analog) are reconstructed at successive time points. Conformations are characterized by two parameters (q and ϕ) that describe continuous transitions among planar, twist, and envelope conformations. Experimentally observed distributions of these parameters agree with those of MC simulations that account for the critical Casimir interactions between particle patches. This system allows for real-time, real-space observation of ring-opening reactions in colloidal cyclopentane catalyzed by a hydrophobic substrate. In this way, colloidal molecules with directional bonds of tunable strength provide a classical analog by which to model chemical transformations.

The experiments, analysis, and modeling are of high quality and demonstrate remarkable control over the direction and strength of colloidal interactions. The agreement between experiments and MC simulations suggest that the system energy is well described by the particle geometry, critical Casimir interactions, and gravity. The potential for investigating catalytic reaction pathways and reactive intermediates using this experimental model is intriguing.

For these reasons, I recommend publication of this work in Nature Communications. I would appreciate it, however, if the Authors could address the following question.

Thank you for your positive words and appreciation.

1) The demonstration of heterogeneous catalysis is quite interesting. Can the Authors comment further on the reaction pathway and the associated energy landscape? Using the reconstructed configurations and the model energy function, it should be possible to construct the energy vs. reaction coordinate diagram and/or identify the transition state. An expanded discussion of this particular example could be helpful in illustrating the potential role of colloidal models in studying “the geometric effects of catalysts”.

This is a very interesting point brought up by the reviewer, which we are currently addressing in more detail using Molecular dynamics simulations. The experimental data does indeed already give some insight into the energies along the reaction path, as shown in Fig. A4 below. Here, we have calculated the bending energy from the bond angles assuming a Hookean bending potential, with a spring constant as determined in the SI. However, this data is preliminary: to get sufficient time resolution, we have used two-dimensional snapshots here to reconstruct the particle positions in the horizontal plane, assuming that their z-coordinates remain unchanged. Latter may be a reasonable approximation for a surface-bound ring, whose particles have one patch attached to the glass, yet it is ultimately not accurate, especially when the ring breaks. Nevertheless, it does offer some insight into the variations in the bending energy of the surface-bound ring upon breaking: Ring breaking occurs for $t > 0$, upon which a large energy barrier is overcome, which we associate with the relaxation of the ring to a low bending energy configuration. The relaxation takes roughly 10 seconds, limited by the diffusion time of the particles moving along the surface, which should set the characteristic time here. While Fig. A4 gives an interesting picture of a possible energy path, we’d like to stress that this (at best) is just the bending energy part; to get the full free energy evolution, one would need the additional entropic contribution, which likely increases significantly upon breaking of the ring. To be

Fig. A4 Energy trace of surface-bound colloidal cyclopentane around breaking

The trace shows large energy variations of the ring, and a particularly large barrier when the ring breaks ($t > 0$ s).

more accurate and include the entropic component, we are currently setting up Molecular Dynamics simulations that should give a more detailed picture of these transition states in a forthcoming publication. To nevertheless implement the referee's idea into the current paper, we have added the sentence:

Changes to the manuscript:

“Fast confocal microscopy imaging can then give detailed insight into energies along the reaction coordinate, identifying the transition states in these catalytic processes.”

Minor point:

2) Figure 2. Captions for (E) and (D) are reversed.

Thank you, we have corrected this issue.

Reviewer #3 (Remarks to the Author):

The authors present a model system of colloidal molecules using a novel fabrication method based on the assembly of single Polystyrene (PS) colloidal patchy particles via Casimir forces. The formation of the various colloidal alkene structures, such as butane, 2-butyne, and cyclopentane, depends on the patch number and orientation on the PS surface (e.g. dimers and tetramers) and how this interact. In particular, the authors visualize the bonding arrangements of cyclopentane using confocal microscopy and characterize the pseudorotation distinguishing the amplitude and the phase, which shows an increasing number of configurations. They support their experimental results with a simulation at the colloidal scale and comparing it with the analog molecular structure. Moreover, they also explore the possible effect of the interaction with the surface on the ring-opening as an analogy for the reactivity of such molecules in the presence of catalyzers. All the results are presented in a very clear and coherent manner. This is an interesting work that shows the vast possibilities of these new colloidal structures to be used as a model for molecular structures to elucidate transition states and specific kinetics.

Thank you for your kind words and positive judgement.

However, there are some comments and issues that need to be addressed before I can recommend it for publication:

1. *From a fabrication point of view, what are the final structure populations? In particular, could the authors elaborate more on the count of cyclopentane structures, as well as for the counts in events?*

We thank the referee for motivating us to look into this. We have performed additional experiments, and determined structure populations following his suggestion. The central graph, incorporated as Fig. 1H in the main text is shown for convenience below (Fig. A5). Here, cluster size distributions are shown for all structures (violet data and left axis) as well as for cyclic structures only (orange data and right axis). Clearly, the number of clusters decreases with cluster size. Interestingly, and in contrast to the population of all structures, the population of cyclic structures does show a clear preference for

Fig. A5 Probability of occurrence of colloidal molecules as a function of their size.

Purple data (left axis) shows all molecules, while orange data (right axis) shows cyclic molecules only. Clearly, larger molecules are increasingly rare, while among the cyclic structures, 5-particle compounds such as cyclopentane are most frequently observed.

certain cluster sizes. Obviously, 5-particle rings (the cyclopentane studied in the manuscript) are most prominent, while smaller rings of 3 or four particles are rare. This can be explained by the (in)compatibility of certain cyclic structures with the tetrahedral bond angles set by the tetrapatch particle geometry. However, this would favour $n = 6$ rings (cyclohexane) and larger rings as well. The fact that we do not observe them as much is likely of kinetic origin (see e.g. Noya *et al.*, Assembly of clath rates from tetrahedral patchy colloids with narrow patches, *J. Chem. Phys.* 151, 094502 (2019)). In addition to Fig. A5, we have also included more details on the kinetic evolution of the clusters in the SI, see Fig. A2 above. This figure shows the continuous growth of clusters, and also indicates some preference for specific cluster sizes in the later stages.

Changes to the manuscript:

Following the referee's comment, we have included cluster size distributions (new Fig. 1H) in the manuscript, together with the following text: "We show size distributions of colloidal molecules in Figure 1H. Clearly, smaller structures are in the majority, but a significant population of larger structures is present in the sample. The population decreases exponentially with size, as predicted for patchy particle systems [19]. Cyclic molecules, however, show a clear preference for a certain number of particles, reflecting their compatibility with the tetrahedral bond angle. Colloidal cyclobutane (Figure 1F) is rarely encountered in our samples. This is not surprising considering its highly strained bond angles. In this configuration, two bonded neighbours make an angle of 90° , far from the ideal angle of 109.5° , causing high bond strain. For atomic cyclobutane, this high bond strain is known to cause the enhanced reactivity of cyclobutane compared to butane, making it much less stable than cyclopentane and cyclohexane that exhibit bond angles much closer to 109.5° . Indeed, we find that colloidal cyclopentane (Figure 1E) is much more ubiquitous in the sample, and by far the most observed colloidal ring structure. Its bonds are much closer to the ideal 109.5° tetrahedral bond angle, compared to cyclobutane. Curiously, six-membered rings - cyclohexanes - are much less frequently observed, though they have a lower bond angle strain, which is likely of kinetic origin [7]"

We also added a new section in the SI relating to Fig. A2 above, to detail the dynamic evolution of the cluster population (section "Supplementary Note 1: Patchy Particle Assembly" and corresponding Supplementary Figure 3.)

2. *There is an inconsistency in the PS particle diameter: in the main text, the authors state that the PS particle has a diameter of 3.7 μm , while in the Materials and Methods they say 2.2 μm . This information might be crucial, as the ratio between PS and patch size might play a crucial role during the molecule fabrication. It would also be useful to have information about the polydispersity of the patches.*

The two mentioned sizes, 3.7 and 2.2 μm , refer to different particles, we apologize if this has not been clear enough. Our patchy particles are produced by fusing polystyrene particles with TPM droplets. The 2.2 μm diameter in the SI refers to the original PS particles before the colloidal fusion process, thus to the precursor of the patchy particle, while the 3.7 μm is the final diameter of the 4-patch particle resulting from the fusion process. We have clarified this in the SI. We have also added particle polydispersities, and in response to comment 4 of the reviewer, their comparison with the interparticle distances.

Changes to the manuscript

We added particle sizes and polydispersities, and comparison with the interparticle distances in the Supplementary Methods as follows: “The patchy particles dimensions were determined using Atomic Force Microscopy (Supplementary Table 1) and confirmed by observation of inter-particle distances of bonded tetramer particles (Supplementary Figure 1). Very good consistency is observed: the mean of the inter-particle distance distribution at 3.8 μm reflects the particle size (particle diameter 3.7 μm) plus twice the patch height ($2 \times 48 \text{ nm} = 96 \text{ nm}$), plus the (short) interaction range, while the standard deviation of $\sigma = 0.11 \mu\text{m}$ reflects the estimated particle size variation (100 nm) and twice the variation of the patch height ($2 \times 5 \text{ nm} = 10 \text{ nm}$).”

3. *Due to the size of the particles, I would expect sedimentation during the alkene assembly step. Could the authors give more experimental information on how the process of assembling the alkenes is done? Is stirring a key parameter to avoid sedimentation and ensure good mixing and therefore interaction of the various patchy particles?*

In most of the experiments reported in the manuscript, we let the particles sediment to the bottom of the sample cell first, before we switch on the critical Casimir attraction to assemble the structures. The structures then grow essentially by two-dimensional diffusion in the plane. No mixing is necessary. The gravitational force has some influence on the observed puckering amplitude, essentially flattening our ring somewhat (see Supplementary Note 4 for a more involved discussion).

Only for the catalytic experiments (last part of the manuscript) we assemble the alkenes in the bulk. This is because in these experiments, the walls are treated to exhibit strong particle-wall interaction. To investigate their “catalytic effect”, we assemble the particles in the bulk, and then let the entire assembled structure sediment to the bottom, so that after contact with the wall, the “catalytic” process can begin. This is done by first letting the particles (without critical Casimir force) sediment on one side of the capillary, turning the capillary upside-down, and then switch on the Casimir interaction so that rings assemble in the bulk (while sedimenting to the other side of the capillary). This process works well, and allows us to study the catalytic effect on cyclopentane rings that have assembled in the bulk. We do not want to additionally stir, in order to not additionally disturb the assembly process.

Changes to the manuscript:

We have added the above details on the procedure in the Methods section of the manuscript as follows: “Particles are left to sediment to the bottom of the sample at room temperature before measurements. We choose a particle concentration such that particles cover between 10 and 15% of the surface after sedimentation. We then switch on the critical Casimir attraction to assemble the structures.”

And somewhat later: “The structures then grow by two-dimensional diffusion in the plane. No mixing is necessary.”

And finally: “For the catalytic experiments we assemble the alkenes in the bulk. This is because in these experiments, the walls are treated to exhibit strong particle-wall interaction. To investigate their “catalytic effect”, we assemble the particles in the bulk, and then let the entire assembled structure sediment to the bottom, so that after contact with the wall, the catalytic process can begin. This is done by first letting the particles (without critical Casimir force) sediment on one side of the capillary, turning the capillary upside-down, and then switch on the Casimir interaction so that rings form in the bulk (while sedimenting to the other side of the capillary). This process allows us to study the catalytic effect on cyclopentane rings that have assembled in the bulk.”

4. From Fig. 1, it is not obvious that the distance from each patchy particle might be constant. Could the authors provide more information on this? The possible polydispersity of the patches (see comment #2), might affect the interparticle distance, and most probably in the case of cyclopentane the reconfiguration events.

We agree that the polydispersity of the patches affect the interparticle distance, and possibly also the reconfigurations of the cyclopentane ring. For our particles, we refer the reviewer to Supplementary Table 1 in the SI, where we show the patchy particle dimensions as determined by AFM, including an estimate for the patch size and polydispersity. To see how these numbers compare to the interparticle distance, we have determined the distribution of particle center-to-center distances as shown in Fig. A6 below. Very good consistency is observed: the mean of the distribution at $\sim 3.8\mu\text{m}$ reflects the particle size ($3.7\mu\text{m}$) plus twice the patch height ($2 \times 48\text{nm} = 96\text{nm}$), plus the (short) interaction range, while the standard deviation of $\sigma = 0.11\mu\text{m}$ reflects the estimated particle size variation (100nm) and twice the variation of the patch height ($2 \times 5\text{nm} = 10\text{nm}$). We have included Fig. A6, and the correspondence to the particle size and polydispersity in the SI. We thank the referee for pointing out the link.

We finally note that the relative sizes of the patch and particle matrix is also of importance for the eventual structure of the molecule. A more detailed discussion can be found in Gong *et al.*, *Nature* (2017).

Fig. A6 Distribution of center-to-center distances of the assembled 4-patch particles. The mean reflects the particle size ($3.7\mu\text{m}$) and twice the patch height ($2 \times 48\text{nm} = 96\text{nm}$), plus the (short) interaction range. The standard deviation $\sigma = 0.11\mu\text{m}$ is consistent with the estimated variation in particle size (100nm) and variation in twice the patch height ($2 \times 5\text{nm} = 10\text{nm}$).

Changes to the manuscript:

We added the graph above and the following text in the SI, Supplementary Methods (see also our reply above): “The patchy particles dimensions were determined using Atomic Force Microscopy (Supplementary Table 1) and confirmed by observation of inter-particle distances of bonded tetramer particles (Supplementary Figure 1). Very good consistency is observed: the mean of the inter-particle distance distribution at $3.8 \mu\text{m}$ reflects the particle size (particle diameter $3.7 \mu\text{m}$) plus twice the patch height ($2 \times 48 \text{ nm} = 96 \text{ nm}$), plus the (short) interaction range, while the standard deviation of $\sigma = 0.11 \mu\text{m}$ reflects the estimated particle size variation (100 nm) and twice the variation of the patch height ($2 \times 5 \text{ nm} = 10 \text{ nm}$).”

5. The authors argue that the shift of the distribution of puckering amplitude to higher values of q for molecules is due to the steric repulsion of H-H. This raises a great experimental challenge, as one could aim to achieve such a steric repulsion in some way. Have the authors considered the presence of the patches somehow in the simulations?

We have considered ways of mimicking an H-atom and the steric repulsion it brings in our colloidal analogues. An obvious solution would be the addition of 1-patch particles which can cap any remaining open patches. However, since there is (currently) no real way of controlling which particle binds where, this would indeed be quite an experimental challenge. A possible way forward is via the precise tuning of the relative attraction magnitudes (via precisely tailored surface wetting properties controlling the crit. Casimir amplitude), thus creating actual balance between attraction-dependent equilibrium states, mimicking the atomic molecule in more detail. We have done this for mixtures of spherical particles A and B (with interaction energies $u_{AA} < u_{AB} < u_{BB}$) where we could tune their phase diagram, but trying to do this on patchy particles would be challenging, though possibly worth exploring. If the reviewer has any other ideas, we'd be happy to hear!

To answer the second part of the question: yes, we have considered the presence of the non-bonding patches in our simulations. They are exactly the same as the bonding ones, and if any more particles are present in the simulation box they can, and do sometimes, bond to the structure. Due to the short range of the critical Casimir force, however, the attraction has only a small influence on our molecules when nothing is bonded.

6. The fact that MgSO_4 is a multivalent salt results in a higher ionic strength of the solution in comparison to using the same molarities for KCl or CaCl_2 . I wonder if the authors could elaborate a bit more on this and on the clear effect on the colloidal molecules assembly.

The salt in this system is there to screen particle charges; thus the first aim is to reduce the electrostatic repulsion so that critical Casimir attraction and electrostatic repulsion can balance. In addition, besides the mere ionic strength, there is another difference in behavior when using different salts, as we discuss in the SI, Supplementary Methods. Addition of MgSO_4 allows us to enhance the attraction contrast between the particle patches and matrix. This is different for other salts: preliminary tests with other ions with 2 or more charges do not show the same effect as magnesium sulfate. Similar effects have been observed before, as referenced in the SI: changing the critical Casimir attraction is sometimes observed when adding different salts (at the same ionic strength).

A deeper reason may be in the special properties of water, specifically how salts structure the water around themselves. In biochemistry, the so-called Hofmeister series is well known to influence how proteins behave, although the effect is not entirely understood. The idea is that different ions structure water around themselves differently, causing a range of secondary effects: different surface tension, different protein solubility, etc. It is not clear that such an effect is at play here, but it does

illustrate that there is more to salt ions than just their charge. Especially in a system like ours, close to the critical point of de-mixing, where such effects may be amplified.

Another effect that could play a role here is the solvent preference of the individual ions. Some salts are “antagonistic”: they consist of a hydrophobic and a hydrophilic ion. It is known that the preference of these ions for either phase may be a driving force for a different water-lutidine phase diagram. In the ‘simple’ magnesium sulfate salt we use, this effect is likely small, but it could play a role, see e.g. Glende et al., “The Vanishing water/oil interface in the presence of antagonistic salt. J. Chem. Phys. 152”, 124707 (2020).

In our work, we use the effect to enhance the contrast between particle patch and matrix, enabling good control over the patch-to-patch bonding.

7. I have my reservations towards the claim of using the surface as an analog for a catalyst. Could the authors prove that the interaction with the hydrophobic surface is strong enough to break a 15 kBT bond between the patches? What is the treatment done on the surface to achieve such hydrophobicity?

The surface treatment is detailed in the Supplementary Methods, and consists of a gas silanization step. The glass becomes hydrophobic, and thus attractive to particle patches via the critical Casimir force. We know from follow-up work of surface-bound 4-patch particles forming two-dimensional crystals (analog of graphene), that the attraction to the glass is quite strong, strong enough to keep the particles attached (while being fully mobile along the surface), even at a ΔT , where the bonds between particles already break. This makes sense: as the surface is specifically hydrophobically treated, we expect that its interaction with a patch is somewhat stronger than that with another patch. On top of this, the flat shape of the substrate will result in stronger interaction with the spherically shaped patch compared to two spherical shapes. (i.e. the higher the curvature of the objects, the lower the effective interaction strength, based on the Derjaguin approximation, see e.g. Lee R White, Journal of Colloid and Interface Science (1983)). Experimentally, we observe that, upon heating the sample, particles start sticking to the glass surface first, *before* patches start sticking to each other. This means that at any given temperature, the attraction particles feel to the glass is always stronger than the attraction towards each other.

Reviewer #4 (Remarks to the Author):

I read the manuscript “Revealing pseudorotation and ring-opening reactions in colloidal organic molecules” by P. J. M. Swinkels and co-workers. The authors harness critical Casimir forces to fabricate clusters of microparticles that mimic the architecture of polyatomic molecules. The valency is implemented using colloids equipped with sticky patches, whereas the floppiness of the bonds stems from the small magnitude of the attractive forces. After showcasing few examples of ‘colloidal molecules’ that can be assembled using this method, the authors focus their attention on the dynamics of cyclopentane, finding strong similarities (and differences) with the puckering configurations of their molecular counterparts.

I find the results presented in the manuscript novel and interesting. As correctly pointed out in the Introduction, the literature is rich of ‘colloidal atoms’ but there are only few examples of colloidal molecules because directional bonds are not easy to implement. In particular, I am not aware of any existing work in which cyclic colloidal molecules have been achieved experimentally. On the other hand, I believe that the manuscript could be more transparent as important pieces of information are now hidden behind a nice story. My main remarks are listed below, in descending order of importance.

1) I find the title, abstract and introduction partly misleading and overstating. While it is claimed that

“colloidal alkanes assembled from tetrameric patchy particles undergo the same chemical transformations as their atomic counterparts”, this is only shown (nicely) for colloidal cyclopentane. There is no result concerning the other molecules shown in Fig. 1.

We have added significantly more data on the behavior of other colloidal molecules, see the additional graphs in both manuscript and Supplementary information (figures A1-A3, and A5 above in this reply), giving a broader picture of colloidal alkanes. This relates to both the puckering and angle distributions (Fig. A1 and A2), as well as the growth (Fig. A3) and structure populations (Fig. A5). In addition to the new data on other molecules, we have specified the sentence in the abstract to more precisely reflect what is shown in the manuscript.

Changes to the manuscript

We have added more data on other colloidal alkanes, see new Figs. 1H, 2C, the Supplementary Figure 3, Supplementary Figure 4 in the manuscript and Supplementary information. We have also specified the most investigated alkene in the abstract as follows: “Here, we show that colloidal alkanes, **specifically colloidal cyclopentane**, assembled from tetrameric patchy particles undergo the same chemical transformations as their atomic counterparts, allowing their dynamics to be studied in real time.”

2) It is unclear to me what the authors exactly see while setting up an experiment. If I understand correctly, they prepare a suspension and the particles self-assemble into ‘molecules’ when they get close to contact (at $T \approx T_{cx}$). Hence, there is little control over the final cluster(s) beyond the binding sites. It is mentioned that some molecules are “rarely encountered”, whereas “cyclopentane is by far the most observed colloidal ring structure”. It would be useful to understand a bit more the statistics and how it evolves. This is particularly important since the molecules have open bonds and clusters will, therefore, grow with time. On the other hand, do they ever spontaneously break?

We have performed additional experiments to elucidate the assembly process and statistics of observed structures. The results, both on the statistics and the kinetics of the assemblies are shown in Figs. A3 and A5 above. We have used this to elaborate on how the structures evolve when setting up the experiment. Fig. A3 (Supplementary Figure 3 in the revised paper) particularly gives an impression of what we see when running the experiment. The real-space snapshot in Fig. A3a directly shows the particle clusters as we see them in bright-field microscopy (in addition, we also focus on individual colloidal structures using 3D confocal microscopy to directly image the fluorescent patches to reconstruct their 3D configuration). Furthermore, a typical assembly process of a colloidal alkane molecule is depicted in Fig. A3b. The evolution of the cluster population is shown in Fig. A3c and d, while the statistics in an advanced stage of assembly is shown in Fig. A5. Latter particularly shows that some cyclic molecules (such as $n = 4$, cyclobutene) are rarely encountered, while cyclopentane ($n = 5$) is by far the most observed colloidal ring structure, as seen by the clear peak at cluster size 5. Spontaneous breakage of the structures is observed at the current attractive strength, as shown in Fig. 4A of the manuscript.

Changes to the manuscript

We have added more data on the assembly of the structures, as well as their statistics, see new Figs. 1H, 2C, Supplementary Figure 3, Supplementary Figure 4 in the manuscript and SI, together with corresponding text in the manuscript and SI.

3) Another relevant issue is the particles-substrate affinity. Near the critical point I expect the Casimir interactions between the colloids and substrate to come also into play. This effect is discussed in relation to Fig. 4, but not considered before. Why? The fact that the substrate is not functionalised (as

in Fig. 4) is not enough to guarantee that critical Casimir interactions do not exist. Do the authors observe free rotational diffusion of single particles? This would indicate that these particle-substrate interactions are somewhat negligible.

For the untreated surface, the particle-wall interaction is small, significantly smaller than the gravitational force. The (untreated) glass is somewhat hydrophilic, while the patches are hydrophobic. Therefore, any expected attraction occurs between particle bulk and the wall. However, as can be seen in Supplementary Figure 2B of the SI, in the phase diagram, we are quite far away from the region where the particle bulks attract each other (the “sweet spot”). We therefore conclude that attraction to the wall is small. Indeed, this is confirmed by the observation of free rotational diffusion of the particles. To quantify the attraction and surface diffusion of the particles in more detail, we have determined their mean-square displacement (MSD) along the surface as shown in Fig. A7 below, where we compare the MSD at $\Delta T=0.27^\circ\text{C}$, where critical Casimir interactions should be still negligible, with $\Delta T=0.04^\circ\text{C}$, which is the temperature of assembly used in the manuscript. In both cases, the particles clearly exhibit diffusion (MSD with slope 1) along the substrate, while the diffusion coefficient decreases slightly at $\Delta T=0.04^\circ\text{C}$. We associate this decrease with a (slight) additional attraction to the substrate, and an increase of the solvent viscosity close to criticality.

Thus, the effect is twofold: Firstly, the viscosity changes close to T_{cx} . Secondly, there is a change in the height of the particle above the surface due to the slight particle bulk-surface Casimir attraction, changing the hydrodynamic interaction of the particle with the surface. To quantify the latter, we use a theoretical model for colloidal diffusion near a rigid wall. With the model we can determine the contribution of the height change of the particles to the diffusion constant. This theoretical prediction is experimentally verified, see for nice explanations Carbajal-Tinoco *et al.*, *Phys. Rev. Lett.* (2007) or Lisicki *et al.*, *Soft Matter* (2014).

The increase in viscosity of the water-lutidine solvent going from $\Delta T=0.27^\circ\text{C}$ to $\Delta T=0.04^\circ\text{C}$ can be estimated to be about 25% close to the coexistence temperature (Mirzaev *et al.*, *Journal of Chemical Physics* (2006)).

Second, the average particle height at $\Delta T=0.27^\circ\text{C}$ (without Casimir attraction) is set by gravity, and we assume it to equal the gravitational height, which from the particle weight, we estimate to be $h_0 = 270\text{nm}$. Using the difference in diffusion coefficients determined from Fig. A7, and the change in

Fig A7 Mean square displacement (MSD) of particles close and far from the coexistence temperature.

In orange, we show the MSD of free particle diffusion at $\Delta T=0.27^\circ\text{C}$. The solid orange line is the associated fit for a diffusion constant of $0.0300 \mu\text{m}^2/\text{s}$. In blue, we show the MSD of free particle diffusion at $\Delta T=0.04^\circ\text{C}$. The solid blue line is the associated fit for a diffusion constant of $0.0205 \mu\text{m}^2/\text{s}$.

viscosity (of 25%, which accounts for a large part of the measured MSD difference), together with the hydrodynamic model, we then obtain an average particle height of $h = 250\text{nm}$ at $\Delta T = 0.04^\circ\text{C}$, thus slightly smaller than without the critical Casimir force. This indicates that there is indeed a slight critical Casimir attraction E_{Cas} between the particle and the surface, but the effect is small, much smaller than the gravitational force. Using the ratio of the two heights, $h/h_0 = \exp(-E_{\text{Cas}}/k_B T)$, we obtain $E_{\text{Cas}} \approx 0.03k_B T$, which is negligible. We thank the reviewer for motivating us to look into that; the surface diffusion measurement could provide a nice tool to measure particle – surface interaction energies for spherical particles, where the hydrodynamic interaction with the wall is known.

4) Page 3, line 76. “... resulting in a binding energy of $\sim 15k_B T$ ”. Where does this number come from?

This number is based on the potentials that are used in our simulations (see Supplementary Note 4 for the details). These potentials have been derived using critical Casimir scaling theory as described in Stuij *et al.* *Soft Matter* (2017), and additionally calibrated by detailed comparison with experimental distributions. We have added a note to clarify this in the main text.

Changes to the manuscript:

We have added the reference and modified the sentence to: “resulting in a **predicted** binding energy of the patches of $\sim 15k_B T$ (see Supplementary Note 4) [17].”

5) Page 3, line 100. “We rapidly acquire image stacks every 10 seconds, roughly 2 times faster than the typical relaxation time of a configuration”. Do all configurations have the same relaxation time?

The two main configurations, the envelope and the twist conformation, theoretically have very similar energies, and hardly any energy barrier exists between the two. In fact, one could argue that they are not really separate conformations at all, but part of a continuous spectrum of conformations (ergo, pseudo-rotation). This implies that all conformations have the same relaxation time, or at least very similar ones.

In Supplementary Figure 5 of the SI, we show the timescale of dynamics. Two timescales are visible: a fast, diffusional timescale, and a slower timescale associated with configurational change of the ring. Consistent with the above (theoretical) argument, there seems to be no other timescale hidden within the correlation decay, and thus we conclude that there is indeed only one relevant timescale for the puckering dynamics.

6) I would spend few words to introduce the puckering amplitude and phase in the main text (not only in the Methods).

Thank you. We have added some more explanation of the puckering amplitude and phase.

Changes to the manuscript:

The section has been modified to: “To analyse the pseudorotation in detail, we determine the puckering amplitude q and phase ϕ from the out-of-plane displacements z_i of the particles. Together, q and ϕ form a polar phase space describing all possible puckering conformations. Given an average plane through the ring, q is a measure of the resulting amplitude of the out-of-plane displacements, while ϕ tells us in what conformation the ring is, as indicated schematically in Fig. 2B [25]. (see Methods for formal definitions of q and ϕ).”

Furthermore, to better explain the transitions in Fig. 2E (2F in the revised manuscript), we have added reference to the schematic in Fig. 2B by adding: “For example, the large phase change from 2 to 3 corresponds to a change from twist to a next-nearest envelope conformation, while that from 3 to 4 corresponds to a transition from twist to envelope, as shown by comparison of the points in Fig. 2F

with the corresponding phase angles in panel (B), from twist (T-1) to envelope (E+2) to twist (T-4).”

7) The puckering amplitude does not depend on gravity. This is an important piece of information, but it is only given at the end the Supplementary file.

Thank you, we have moved this information to the main text, as we agree it is important information.

Changes to the manuscript:

We have added the following sentence when discussing the puckering amplitude: “As shown by simulations in Supplementary Note 4, the peak position depends on the presence of gravity: without gravity, the maximum of the probability density is shifted towards larger values ($q_N \sim 0.15$) compared to the experimental measurement, meaning that the colloidal cyclopentane ring is more puckered in a system without gravity, as expected. Nevertheless, the presence of a finite puckering amplitude is surprising from an energetic point of view, as the flat ring ($q = 0$) has both the lowest bending and gravitational energy.”

Reviewers' Comments:

Reviewer #1:

Remarks to the Author:

I am fully satisfied with the authors' response to my initial comments and with the subsequent revision of the manuscript. I believe that the extra details and data added to the manuscript greatly increase its clarity and I am happy to recommend publication.

Just as a tiny, picky point, in Figure 1B, the authors report a schematic of the compositional fluctuations between two patchy particles and write "Lutidine fluctuation". Lutidine does not fluctuate, but its concentration.

Reviewer #2:

Remarks to the Author:

The Authors have provided a thorough response to questions and suggestions raised by the Reviewers. The edits and additions have further strengthened the clarity of the manuscript and the support of its claims. I recommend publication in Nature Communications.

Reviewer #3:

Remarks to the Author:

The authors have carefully addressed my major and minor comments, supporting their responses with extra experiments when needed and the corresponding analysis. In particular, some interesting results show up when counting the population of the cyclic structures. Moreover, the discussion of the questions was well reasoned and clear. The new additions strengthen the already very consistent and elegant work.

I, therefore, recommend the article for publication, with some minor issues to update:

- 1) Please, indicate the total number of structures counted for the Probability of occurrence of colloidal molecules as a function of their size.
- 2) Do the authors use AFM imaging mode to determine the dimension of the particles? If yes, it would be great to add the corresponding images to the SI.
- 3) Reference [1] incomplete, missing information about volume, issue, page number of the article.

Reviewer #4:

Remarks to the Author:

The authors carefully addressed all concerns of the reviewers, both in their rebuttal letter and in the new version of the manuscript. The additional figures (e.g. Fig 1H) provide a clear understanding of the experimental playground for any researcher wishing to perform similar measurements.

All in all, the work is an important contribution to the field of colloidal self-assembly and a shining example of how microscopic colloids shed light on molecular problems.

REVIEWERS' COMMENTS

Reviewer #1 (Remarks to the Author):

I am fully satisfied with the authors' response to my initial comments and with the subsequent revision of the manuscript. I believe that the extra details and data added to the manuscript greatly increase its clarity and I am happy to recommend publication.

Thank you very much for your very supportive judgement.

Just as a tiny, picky point, in Figure 1B, the authors report a schematic of the compositional fluctuations between two patchy particles and write "Lutidine fluctuation". Lutidine does not fluctuate, but its concentration.

Thank you for pointing this out, we have adapted Figure 1B such that the schematic now reads "lutidine concentration fluctuations".

Reviewer #2 (Remarks to the Author):

The Authors have provided a thorough response to questions and suggestions raised by the Reviewers. The edits and additions have further strengthened the clarity of the manuscript and the support of its claims. I recommend publication in Nature Communications.

Thank you for your positive words and recommendation.

Reviewer #3 (Remarks to the Author):

The authors have carefully addressed my major and minor comments, supporting their responses with extra experiments when needed and the corresponding analysis. In particular, some interesting results show up when counting the population of the cyclic structures. Moreover, the discussion of the questions was well reasoned and clear. The new additions strengthen the already very consistent and elegant work.

Thank you for your kind words and positive judgement.

I, therefore, recommend the article for publication, with some minor issues to update:

1) Please, indicate the total number of structures counted for the Probability of occurrence of colloidal molecules as a function of their size.

We have indicated the total number of structures counted in the caption of Figure 1.

2) Do the authors use AFM imaging mode to determine the dimension of the particles? If yes, it would be great to add the corresponding images to the SI.

We have added a representative example of an AFM measurement to the SI. Supplementary Fig. 1 shows the AFM measurement and the parameters we extract from such measurements. A few lines have been added to the methods section describing the AFM measurements.

3) Reference [1] incomplete, missing information about volume, issue, page number of the article.

We fixed the reference error.

Reviewer #4 (Remarks to the Author):

The authors carefully addressed all concerns of the reviewers, both in their rebuttal letter and in the new version of the manuscript. The additional figures (e.g. Fig 1H) provide a clear understanding of the experimental playground for any researcher wishing to perform similar measurements.

All in all, the work is an important contribution to the field of colloidal self-assembly and a shining example of how microscopic colloids shed light on molecular problems.

We are happy to hear the reviewer has enjoyed our work, and thank them for their supportive comments.